# In Vitro Analysis of Matched Isolates from Localized and Disseminated Gonococcal Infections Suggests That Opa Expression Impacts Clinical Outcome

**DOI:** 10.3390/pathogens11020217

**Published:** 2022-02-07

**Authors:** Cheng-Tai Wu, Po-Wei Huang, Chia-Hsuan Lin, Daniel C. Stein, Wenxia Song, Sung-Pin Tseng, Liang-Chun Wang

**Affiliations:** 1Department of Marine Biotechnology and Resources, National Sun Yat-Sen University, Kaohsiung 804, Taiwan; gcvold@g-mail.nsysu.edu.tw (C.-T.W.); lchchiahsuan@g-mail.nsysu.edu.tw (C.-H.L.); 2Division of Urology, Department of Surgery, Zuoying Branch of Kaohsiung Armed Forces General Hospital, Kaohsiung 813, Taiwan; m870993@gmail.com; 3Department of Cell Biology and Molecular Genetics, University of Maryland, College Park 20742, MD, USA; dcstein@umd.edu (D.C.S.); wenxsong@umd.edu (W.S.); 4Department of Medical Laboratory Science and Biotechnology, Kaohsiung Medical University, Kaohsiung 807, Taiwan; tsengsp@kmu.edu.tw

**Keywords:** *Neisseria gonorrhoeae*, aggregation, local infection, disseminated gonococcal disease, opacity protein, lipooligosaccharide, infection outcome

## Abstract

Gonorrhea is the second most common sexually transmitted infection, which is primarily localized but can be disseminated systemically. The mechanisms by which a localized infection becomes a disseminated infection are unknown. We used five pairs of *Neisseria gonorrhoeae* isolates from the cervix/urethra (localized) and the blood (disseminated) of patients with disseminated gonococcal infection to examine the mechanisms that confine gonococci to the genital tract or enable them to disseminate to the blood. Multilocus sequence analysis found that the local and disseminated isolates from the same patients were isogenic. When culturing in vitro, disseminated isolates aggregated significantly less and transmigrated across a polarized epithelial monolayer more efficiently than localized isolates. While localized cervical isolates transmigrated across epithelial monolayers inefficiently, those transmigrated bacteria self-aggregated less and transmigrated more than cervical isolates but comparably to disseminating isolates. The local cervical isolates recruited the host receptors of gonococcal Opa proteins carcinoembryonic antigen-related cell adhesion molecules (CEACAMs) on epithelial cells. However, the transmigrated cervical isolate and the disseminated blood isolates recruit CEACAMs significantly less often. Our results collectively suggest that switching off the expression of CEACAM-binding Opa(s), which reduces self-aggregation, promotes gonococcal dissemination.

## 1. Introduction

Gonorrhea is a common sexually transmitted infection (STI) caused by the Gram-negative bacterium *Neisseria gonorrhoeae* (GC) [1]. Local genital infections in both men and women can result in pain during urination and urethral discharge. However, infection is often asymptomatic, especially in women [2,3,4], which allows for silent transmission. Infections in women can lead to complications, such as pelvic inflammatory disease (PID) and disseminated gonococcal infection (DGI), resulting in infertility or predisposing women to ectopic pregnancy [5]. Moreover, antibiotic-resistant gonorrhea is a significant public health issue worldwide and an increasing socioeconomic burden [6]. The growing number of ceftriaxone and azithromycin-resistant strains [7,8], combined with asymptomatic and underreported infections [9], highlight the need for understanding mechanisms for gonococcal colonization and progression to dissemination in order to develop new prevention and treatment strategies.

Multiple GC surface molecules, such as pili, opacity-associated protein (Opa), and lipooligosaccharide (LOS), are capable of phase and antigenic variation and have been implicated in GC colonization of various anatomic locations and eliciting different sequelae [10]. The type IV pili, consisting of the major protein PilE and minor protein PilC, is responsible for initiating the interaction of GC with host cells [11,12]. Opa proteins, with eleven variants encoded by eleven different genes, are responsible for intimate interactions with host cells and the invasiveness of GC [13]. However, in the absence of Opa expression, GC is more capable of tissue penetration [14]. GCs also interact with each other through these surface molecules, which may favor colonization [14,15,16]. Opa proteins expressed on one gonococcus adhere to the LOS of neighbor bacteria by lectin-like interactions [17]. Phase and antigenic variations of these surface molecules potentially evade host defense [18]. It has been postulated that the phase variation of these surface molecules [18,19,20] renders GC the capability of infecting the epithelia of various anatomic locations, leading to different outcomes, including asymptomatic, PID, and DGI.

GC isolated from the genital tract (localized) and the blood (disseminated) from patients show several major differences. First, colonies of most disseminated isolates are much less opaque than those from genital tract isolates [21]. Second, disseminated GC shows resistance to normal human serum [22]. Third, the expression of PorB IA correlates with disseminated gonococcal infection [23]. In addition, increased penicillin resistance or special nutrition requirements have also been reported but varied with regional or host factors [21,22,24,25].

Using human cervical tissue explants and isogenic MS11 expressing no or a single Opa isoform, we have previously shown that expression of CEACAM-binding but not heparan sulfate proteoglycan (HSPG)-binding Opa impedes GC penetrating into sub-epithelial tissue [26,27], suggesting the importance of Opa phase variation in clinical outcomes of infection. GC recovered from patients with urogenital, cervical, and rectal infections are mostly Opa+ [28], but GC recovered from patients with DGI are usually Opa negative (Opa−) [29,30]. In human challenge studies, Opa− GC converted to Opa+ and LOS underwent phase variation when recovered from male volunteers after inoculation [31,32]. These studies suggest a relationship between the variable expression of the Opa and LOS and infection outcomes.

While the roles of surface molecules in GC infection have been extensively studied in vitro, how the phase variation of these surface molecules contributes to infection outcomes in vivo has not been well studied. It is unclear how much of what we learned from strains cultured in laboratories for a long time can be extrapolated to understand GC infection in vivo. This study compared GC isolates from the genital tract with those isolated from the blood of the same DGI patients, the MS11 strain that has been cultured in the laboratory for a long time, and MS11 isogenic derivatives expressing Opa that could not phase vary, to relate the variants of surface molecules on GC with clinical outcomes in patients.

## 2. Results

### 2.1. Isogenicity Characterization of Disseminated and Localized Gonococcal Isolates from the Same Patient

To understand how a localized infection progresses to a disseminated infection, GC collected from the cervix or urethra (localized) and the blood (disseminated) from the same patients were used. We determined if the cervical/urethra and blood isolates originated from the same strain in individual patients. We sequenced six housekeeping genes (abcZ, adk, gdh, pdhC, pgm, and aroE) that were previously used in multilocus sequence typing (MLST) for gonococcal phylogeny identification [33,34]. The results showed that these six housekeeping gene sequences were identical between the localized and disseminated isolates of patients 21, 23, 29, and 63, suggesting that the disseminated isolate was derived from the organism causing the local infection (Table 1). Three of the six housekeeping genes in the two isolates from patient 61 showed different sequence types, indicating that patient 61 had a mixed infection [35]. Therefore, this patient was excluded from this study.

### 2.2. Disseminated Isolates Transmigrate across Polarized Epithelial Monolayers More Efficiently Than Localized Isolates

We have previously shown that the GC MS11 strain lacking Opa or expressing HSPG-binding Opa transmigrates across the endocervical epithelium and penetrates the subepithelium much more efficiently than the GC-expressing phase-variable Opa or CEACAM-binding Opa [14]. As penetration allows GC to enter tissues and disseminate to other parts of the host body, we hypothesized that disseminated isolates transmigrate across polarized epithelial monolayers more efficiently than localized isolates. To test this hypothesis, we incubated piliated, localized, and disseminated isolates from female patients 21, 23, and 63 and male patient 29 with polarized T84 monolayers apically. Bacteria collected from the basal chamber were counted as transmigrated ones. We found that the disseminated isolates from all three female patients transmigrated across polarized T84 epithelial cells 10- to 1000-fold more than their isogenic localized isolates (Figure 1a), even though the level of increases varied from patient to patient. However, both localized and disseminated isolates from the male patient showed no detectable transmigration. We further compared their adherence and invasion capabilities using a gentamicin-resistant assay and did not find any significant differences in GC invasiveness or adherence between the disseminated and localized isolates in all patients (Figure 1b,c). These results indicate that disseminated isolates of female patients transmigrate across polarized epithelial monolayers more efficiently than their localized isolates, specifically in an intercellular manner and independent of their ability to adhere or invade T84 cells.

### 2.3. Disseminated Isolates Aggregate Less Than Localized Isolates

Our previous studies have shown that strong GC-GC interactions, which lead to the formation of large aggregates, inhibit GC transmigration [14,26]. We hypothesize that differential aggregation may contribute to the different transmigration efficiencies of disseminated and localized isolates. To examine bacterial aggregation, GC cultures were dispersed with significant vortexing, added to coverslip chambers, incubated for 6 h, and imaged live by a ZEISS Axio Observer microscope. We evaluated aggregation levels by measuring the area of aggregates (Figure 2a–e). We found that the disseminated isolates formed significantly smaller aggregates than localized isolates of each patient (Figure 2f). Notably, both localized and disseminated isolates from the male patient, which failed to transmigrate across the polarized epithelial monolayer, formed large compact aggregates. These data support the notion that locally-colonizing GCs form larger aggregates than disseminated GCs, consistent with the inhibitory effect of GC aggregation on transmigration.

### 2.4. Isoforms of LOS and Opa Expressed by Localized and Disseminated Isolates

Opa and LOS play critical roles in GC aggregation and infection of epithelial cells [14,17,36]. GC isolated from a patient suffering from DGI has been shown to express a truncated LOS [37]. Based on the different transmigration efficiencies and aggregation patterns between the localized and disseminated isolates shown above, we hypothesize that they express different Opa isoforms and/or LOS structures. We isolated and resolved LOS from the pairs of isolates using SDS-PAGE and visualized them by silver stain (Figure 3a). LOS from MS11, MS11ΔlgtD, MS11ΔlgtF, and MS11ΔlgtE with known LOS structures served as controls [38]. LOS molecules from all patient isolates exhibited similar migrating patterns as MS11ΔlgtD, which contain the lacto-*N*-neotetrose structure without the terminal GalNAc attached. Western blotting with the MAbs 6B7 specific for Galβ1-4GlcNAc of lacto-N-neotetrose failed to bind the LOS of 23C (Figure 3b).

We analyzed Opa proteins expressed by these strains by Western blot, probing with the MAb 4B12 binding to Opa (potentially loop-1 and semi-variable region, reference: Developmental Studies Hybridoma Bank, Iowa city, IA, USA). We found that the intensity of Mab binding differed between the strains of each pair, but that 63B and 63C produced different migration profiles (Figure 3c). With the same amount of proteins loaded, the different MAb 4B12 staining densities may be caused by various Opa expression levels or Opa phase or antigenic variation. Taken together, LOS and Opa isoforms from the localized and disseminated isolates can be different in the levels and/or structures.

### 2.5. Transmigrated GC Have Enhanced Transmigration and Reduced Aggregation Ability

To examine how colonizing GC becomes disseminating GC, we chose the localized cervical isolate from patient 63 (63C) that expressed isogenic PorB, nhba, and PilE that were expressed by disseminated isolate (63B) bacteria (Appendix A). We collected the cervical isolate GC that transmigrated across polarized epithelial monolayers after each of the four rounds of transmigration assays. We found that the transmigration efficiency of the cervical isolate increased after each round of transmigration and became comparable to the blood isolate after four rounds of transmigration (Figure 4a). Even though they adhered to epithelial cells at a similar level (Figure 4b), the transmigration-enhanced cervical isolate aggregated significantly less than the original cervical isolate but similarly to the blood isolate (Figure 4c,d). These data suggest that the transmigration-based selection can drive the localized strain to acquire the phenotype of the disseminated isolate.

### 2.6. LOS and Opa Isoforms Expressed by the Transmigration-Enhanced Cervical Isolate

Because LOS and Opa can undergo phase variation, we examined LOS and Opa expressed by the transmigration-enhanced strains. We found that LOS from the cervical, blood, and transmigration-enhanced cervical isolates exhibited similar migrating patterns as MS11ΔlgtD, which contains the lacto-*N*-neotetrose structure without the terminal GalNAc attached (Figure 5a). Western blotting was used to further confirm the lacto-*N*-neotetrose structure in these isolates, probing with the MAbs 6B7 specific for Galβ1-4GlcNAc of lacto-*N*-neotetrose. We found that all strains were positively stained by the MAb, indicating that they shared the lacto-*N*-neotetrose structure, identical with MS11ΔlgtD (Figure 5b). These results suggest that the LOS structure remains the same during the GC switching from localized colonization into disseminating mode.

We compared Opa proteins expressed by the localized isolate 63C before and after four rounds of transmigration using Western blot, probing with the MAb 4B12. MS11ΔOpa and isogenic MS11 expressing OpaH that binds to CEACAMs (Opa_CEA_) were used as controls. We found that before transmigration, the 63C isolate showed two bands with slightly different molecular weights than Opa_CEA_ expressed by MS11 (Figure 6a). After four rounds of transmigration, the transmigration-enhanced cervical isolate 63CT4 showed a single band with strong MAb 4B12 staining and faster migration than 63C but similar to the disseminated isolate 63B. This result suggests that the transmigration-enhanced cervical isolate 63CT4 expresses Opas differently from the cervical isolate 63C before transmigration. To test whether the different Opas expressed by 63CT4 and 63C were attributed to phase variation or intramolecular gene recombination between the various Opa genes, we performed Opa gene fingerprint analysis. We designed PCR primers specific to the conserved regions of *opa* genes, which enabled us to amplify the hyper-variable regions 1 and 2 of *opa* genes (Figure 6b). PCR products were digested with the restriction enzymes TaqI and HpaII [39,40]. The DNA fragment patterns of 63CT4 and 63C obtained after digestion with TaqI (Figure 6c) or HpaII (Figure 6d) appeared to be the same, suggesting that the genomic organization of the *opa* genes in 63CT4 did not undergo recombination compared to 63C. Taken together, our results suggest that the Opa phase-variation, but not LOS phase variation, contributes to the infection mode transition from localized colonization to dissemination.

### 2.7. Loss of CEACAM-Binding Opa Expression Impacts Dissemination

Opa proteins have been divided into two groups, based on their host receptors as CEACAMs or HSPG [41,42]. Most previous studies have focused on FA1090 and MS11 [13]. However, allelic diversity in *opa* genes has been observed [43]. We have previously shown that Opa_CEA_ expression inhibits GC transmigration across the endocervical epithelium, which led to our hypothesis that the disseminated isolates reduce Opa_CEA_ expression compared to localized isolates. To test this hypothesis, we compared the ability of the cervical and blood isolates to recruit CEACAMs, as an indication of Opa_CEA_ expression. The localized cervical isolate 63C and the transmigration-enhanced isolate 63CT4 were inoculated from the apical surface of polarized T84 cell monolayers for 6 h. GC, CEACAMs (CEA), and cell-cell junctional protein E-cadherin (E-cad) were visualized by antibody staining and analyzed using three-dimensional immunofluorescent microscopy. MS11 expressing no or non-phase variable Opa_CEA_ served as controls. We evaluated the level of CEACAM recruitment to GC aggregations by measuring the percentage of GC aggregations with CEACAM staining concentrated beneath (Figure 7a). About 75% of the cervical isolate 63C aggregates recruited CEACAMs, similar to the MS11 Opa_CEA_ control. Furthermore, the percentage of 63C aggregates recruiting CEACAMs was significantly higher than those of both the blood isolate 63B and the transmigration-enhanced isolate 63CT4 (Figure 7b–g). This result indicates an association of GC dissemination with a reduced Opa_CEA_ expression.

We have previously shown that CEACAM-Opa interactions stabilize the apical junction, promoting GC colonization and inhibiting GC penetration in the human cervix [26,44]. To further confirm the differential expression of Opa_CEA_ between the localized and disseminated isolates, we compared their effects on the distribution of E-cadherin. We quantified the translocation of E-cadherin from the cell-cell junction to the cytoplasm (Figure 7b–f) using the fluorescent intensity ratios (FIR) of E-cadherin at the cytoplasm to the junction region (Figure 7h) [14]. We found that the FIR of E-cadherin at the cytoplasm to the junction was higher in epithelial cells inoculated with blood (63B) and transmigration-enhanced isolate (63CT4) than those inoculated with the cervical isolate 63C. Notably, the E-cadherin cytoplasm at the junction FIR in epithelial cells inoculated with the transmigration-enhanced cervical isolate 63CT4 significantly increased compared to those inoculated with Opa_CEA_ GC (Figure 7i). These results suggest that the cervical isolate expresses a higher level of Opa_CEA_ than the blood and transmigration-enhanced isolates, resulting in E-cadherin stabilization and localized infection. Collectively, our results suggest that GC switch from local colonization mode into disseminating bacteria by turning off or reducing the expression of Opa_CEA_.

## 3. Discussion

While 1.6 million cases of gonorrhea were reported in 2018 in the United States [45], less than 1% of these individuals developed disseminated infection [46]. It remains unclear why the incidence of DGI is low. We employed pairs of GC isolates from the cervix/urethra and the blood of the same patient to search for determinants that allow GC to switch from localized colonization into disseminating mode. This study links GC in vitro phenotypes and surface molecules with clinical outcomes. By selecting transmigrated cervical isolates, an in vitro process driving the switch of GC from colonization to dissemination model, which enables us to demonstrate an association of the low self-aggregation and high transmigration phenotypes with disseminated infection as the clinical outcome. We further identified that the loss or reduced expression of CEACAM-binding Opa, but not LOS phase variation, could explain the local dissemination switch in one of these strains. Thus, Opa phase variation potentially determines whether a GC infection progresses from localized to disseminated infection in the clinical setting.

Previous studies suggest an essential role for Opa and LOS in GC infectivity and host responses [14,17,26,47,48,49,50]. Phase variation of these surface molecules has been implied for the progression of the GC infections. Using pairs of localized and disseminated isolates from a few DGI patients, this study suggests that Opa phase-variation contributes to clinical outcomes. Lower or loss of expression of Opa_CEA_ may allow for the switch from the localized infection into a disseminated infection. We have previously shown that MS11 Opa_CEA_ blocks penetration into the endocervical subepithelium using cervical explants [26]. Here, we show that the disseminated isolates transmigrate and disrupt the apical junction more efficiently than the localized cervical isolates, suggesting that transmigration across the epithelium is the mechanism by which GC disseminates into the blood.

The terminal lacto-*N*-neotetrose of LOS has been shown to be important to GC invasion into epithelial cells [50]. Here, we found that LOS in both disseminated blood and localized cervical isolates, and transmigration-enhanced isolate from the same patient, have a similar molecular weight and the lacto-*N*-neotetrose structure, using MS11ΔLgtD LOS containing lacto-*N*-neotetrose as a control. Previous studies show that F62 WT and ΔLgtD mutant display similar invasion and adherence efficiency in ME180 cells [50]. Here, we also did not detect any significant difference in the invasion of all the isolates using gentamicin-resistant assay. These results together support the notion that the lacto-*N*-neotetrose structure of LOS determines GC invasion efficiency. The conservation of the LOS structure between blood and cervical isolate indicates that the enhanced transmigration of the disseminated isolates is not dependent on LOS phase variation. However, survival in the blood is enhanced by the expression of lacto-*N*-neotetrose as it can be sialylated [51].

Opa-LOS interactions between neighboring bacteria lead to GC aggregation. GC aggregation likely also regulates GC interaction with epithelial cells, consequently influencing GC colonization and transmigration. We have previously shown that GC expressing different Opa and/or LOS isoforms impact the size and compactness of aggregation on abiotic surfaces or the surface of epithelial cells [52]. This study shows that the localized isolates aggregate and recruit CEACAMs more than the disseminated isolates, supporting that Opa-dependent aggregation promotes colonization. Colonization requires GC adherence to epithelial cells and survival in the presence of antimicrobial peptides secreted by the female reproductive tract. We have previously found that GC aggregation enhances GC’s ability to resist antimicrobials, probably by altering the penetration efficiency of antimicrobials into aggregations [52,53]. We speculate that the localized cervical GC may be sustained in the female reproductive tract better than disseminated GC through enhanced aggregation and antibiotic resistance.

In addition to phase variation, Opa proteins also undergo random antigenic variation, and consequently, Opa proteins expressed by isolates from different patients can vary in their sequences [54]. Our fingerprint analysis of the HV1 and HV2 regions of *opa* genes between a localized and transmigration-enhanced isolate did not detect sequence variations. However, such an assay cannot include the possibility of phase and antigenic variations. Importantly, our CEACAM recruitment assay reveals that the local cervical isolate recruits CEACAMs to adherent sites on the apical surface of epithelial cells much more than the disseminated isolate, which functionally showed that localized isolates express a higher fraction of Opa_CEA_ than the disseminated isolate. Nonetheless, it remains unknown if the disseminated isolate decreases the expression of Opa_CEA_ and/or increases Opa_HSPG_ expression.

Our results strongly support a critical role of Opa phase variation in determining the clinical outcomes of GC infection, disseminated, or localized infection. However, we cannot exclude possible contributions from the variation of other GC surface molecules, such as pili and porin. Pili can also undergo phase and antigenic variation and have been shown to be essential for initiation of GC infection, GC adherence, invasion, and transmigration [55,56,57]. Porin, a hydrophilic channel protein, that comprises 60% of GC outer membrane protein, has been found to mediate invasion in Chang cells [58]. The contribution of pili and porin to GC infection progression from localized to disseminated is our future interest.

Using patient isolates, this study provides evidence that GC can switch infection modes from local to dissemination by changing Opa expression. However, localized and disseminated infections may not be solely controlled by Opa variation. Differences in host infection sites can contribute to clinical outcomes. Previous studies have suggested that age, ethnicity, and certain genetic traits are associated with the contract rate [59,60]. Moreover, the use of primary urogenital epithelial cells and/or the tissue explant may generate a more clinical-relevant result. Studies on the relationship of GC isolates and patients’ physiological condition would help further understand the mechanisms by which localized GC infection progress into a disseminated infection. Our data would suggest that targeting Opa expression might enhance the prevalence of DGI, and any potential gonococcal vaccine should not contain Opa protein.

## 4. Materials and Methods

### 4.1. Bacteria Strains

*N. gonorrhoeae* strain MS11 that expressed phase variable Opa and pili (MS11Opa+ Pil+), MS11∆Opa, MS11∆LgtD, MS11∆LgtE, and MS11∆LgtF were previously described [15]. The strain expressing single Opa (MS11Opa_CEA_) was described by Stein et al. [14]. Five sets of strains isolated from blood and cervix or urethra were kindly provided by Dr. Peter Rice as previously described [61]. Piliated GC colonies were identified and used in experiments based on colony morphology using a dissecting light microscope. GCs were grown on plates with GC medium base (BD, Franklin lakes, NJ, USA) and 1% Kellogg’s supplement (GCK) [62] at 37 °C with 5% CO_2_ for 16–18 h before use in experiments.

### 4.2. Epithelial Cells

The human carcinoma cell line T84 (ATCC, Manassas, VA, USA) was maintained in DMEM: Ham F12 (1:1) supplemented with 7% heat-inactivated fetal bovine serum (FBS). Cells were seeded at 6 × 10^4^ per transwell (6.5 mm diameter and 3 μm pore size, Corning, NY, USA) and cultured for ~10 days until the transepithelial electrical resistance (TEER) reached >1000 Ω/transwell. TEER was measured using a Millicell ERS volt-ohm meter (EMD Millipore, Burlington, MA, USA).

### 4.3. Multilocus Sequence Typing

GC was incubated on GCK at 37 °C in 5% CO_2_ for 16–18 h, suspended in ddH_2_O, and boiled for 5 min to generate a genome template. Housekeeping genes were amplified via PCR using a SuperCycler SC-200 (Kyratec Pty Ltd., Brisbane, Australia). The primer sets for house keep genes were designed according to Vidovic et al. [34], which were: abcZ-F(5′-AATCGTTTATGTACCGCAGG-3′), abcZ-R(5′-GAGAACGAGCCGGGATAGGA-3′); adk-F(5′-CGTTCGGCATTCCGCAAATCTCT-3′), and adk-R(5′-CGACTTTGATGTATTTCGGCGC-3′); aroE-F(5′-GATTCATCAGCAATTTGCCCTTCA-3′) and aroE-R(5′-CCGCGCCAGAGGGCGTAGGAAGC-3′); gdh-F(5′-ATGTTCGAGCCGCTGTGGAACAA-3′) and gdh-R(5′-CTTCAACGGCCTTGCCCAAATCC-3′); pdhC-F(5′-GTTCCGGTACGATTCTGCAAGAAG-3′) and pdhC-R(5′-CGGTTTCTTTGCTGACTTTGCCT-3′); pgm-F(5′-GAACACGGCGGAGAAGCCATAATG-3′) and pgm-R(5′-CTTGCGTATCCGCTTCAAAACGCA-3′). Electrophoresis was performed with 1% agarose gel in TAE buffer (Tools biotech Ltd., New Taipei, Taiwan). PCR products were extracted from the gel (Easyprep Gel and PCR extraction kit; Tools biotech Ltd.) and submitted to Sanger sequencing (Mission biotech Ltd., Taipei, Taiwan). The housekeeping gene sequences were aligned with the reported sequence at the PubMLST.org website to obtain the MLST typing number [63].

### 4.4. Adherence, Invasion and Transmigration Assay

The assays were performed as previously published [36]. Briefly, polarized T84 cells were incubated apically with piliated GC (1 × 10^5^/well, MOI ~ 10) at 37 °C in 5% CO_2_ for 3 h for adherence and 6 h for invasion and transmigration assays. For adherence, cells were lysed with PBS containing 1% saponin (Sigma-Aldrich, Burlington, MA, USA) and plated on GCK to determine the number of GC. For transmigration, GC in the basolateral chamber was enumerated as the number of transmigrated GC. For invasion, infected cells were treated with gentamicin, washed, and lysed with PBS containing 1% saponin. The number of invaded GC was determined by plating an aliquot of the lysate on GCK.

### 4.5. Transmigration-Enhanced Cervical Isolate

Transmigration assays were performed on piliated GC cervical isolate. GC that transmigrated were isolated on GCK agar, and six colonies were picked, amplified by passage on GCK agar, and then used for the next round of transmigration; the selection was performed for a total of four rounds. The transmigrated GCs were picked, streaked on a GCK plate, and cryo-preserved for the rest of the experiments.

### 4.6. Opa Genotyping

*N. gonorrhoeae* was incubated on GCK at 37 °C in 5% CO_2_ for 16–18 h, suspended in ddH_2_O, and boiled for 5 min to obtain genome DNA template. PCR was performed by SuperCycler SC-200 (Kyratec Pty Ltd.). The opa typing primer sets: opa-up(5′-GCGATTATTTCAGAAACATCCG-3′) and opa-down(5′-GCTTCGTGGGTTTTGAAGCG-3′) were used [39]. PCR products were purified and digested with Taq1-v2 (New England Biolabs Ltd., Ipswich, MA, USA) and Hpall (New England Biolabs Ltd.) based on the protocol suggested by the manufacturer. Electrophoresis was performed by 5% agarose gel in TAE buffer then imaged by DigiGel-Digital Gel Image system (TOPBIO Ltd., New Taipei, Taiwan).

### 4.7. LOS Purification

Gonococcal LOS were prepared from plated cultures as previously described [64]. GC were suspended, and the absorbance of the GC suspension at 650 nm was adjusted to 0.4. GCs (1.5 mL aliquots) were added to microfuge tubes and centrifuged at 4000× *g* for 5 min at 4 °C. The supernatants were removed, the pellets were resuspended in 50 μL LOS lysis buffer (2% SDS, 4% 2-mercaptoethanol, 10% glycerol, 1 M Tris, pH 6.8, and bromophenol blue), and the solution was heated at 100 °C for 10 min. After cooling, proteinase K was added to the lysates and incubated at 60 °C for 60 min. The lysates were then used as crude LOS extraction and stored at 4 °C.

### 4.8. Silver Stain

LOS lysates were subjected to sodium dodecyl sulfate-polyacrylamide gel electrophoresis (SDS-PAGE) on a 4–15% tris-glycine gradient gel (SMOBIO Technology, Inc., Hsinchu, Taiwan) in tris-glycine running buffer, according to the protocol suggested by the manufacturer. The LOS was visualized by silver staining as described previously [65]. The gels were fixed overnight in 40% ethanol 5% formaldehyde at 4 °C, washed for 5 min twice with ddH_2_O, soaked in 0.02% Na_2_S_2_O_3_ for 1 min, and washed twice with ddH_2_O. The gels were incubated in 0.1% AgNO_3_ for 15 min, rinsed three times with ddH_2_O, and developed in 3% Na_2_CO_3_, 0.05% formaldehyde, 0.0004% Na_2_S_2_O_3_. Development was terminated with 2% acetic acid. Images were acquired using a DigiGel-Digital Gel Image system (TOPBIO Ltd.)

### 4.9. Immunoblotting

GC cell lysates were prepared by culturing agar-grown bacteria in basal medium (DMEM: Ham F12 (1:1), 7% FBS) for 6 h at 37 °C with 5% CO_2_. GCs were collected by centrifugation. For Opa blotting, GC pellets were lysed by boiling bacteria in Laemmli sample buffer [50 mM Tris-HCL pH 6.8, 4% SDS, 10% glycerol, 5% 2-mercaptoethanol, and 0.05% bromophenol blue] for 5 min. Lysates were resolved using Q-PAGE™ TGN Precast Gel (SMOBIO Technology, Inc.) and analyzed by Western blot. Blots were probed with 4B12 antibody (Developmental Studies Hybridoma Bank, IA, USA). For LOS blotting, pellets of 6 × 10^8^ GC were lysed by 50 μL lysing buffer [2% SDS, 4% 2-mercaptoethanol, 10% glycerol, 1 M Tris, pH 6.8, and 0.05% bromophenol blue] and treated with proteinase K. Equal amounts of LOS extractions were resolved using Q-PAGE™ TGN Precast Gel (SMOBIO Technology, Inc.) and analyzed by Western blot. Blots were probed with 6B7 antibody (Developmental Studies Hybridoma Bank, Iowa City, IA, USA). Images were acquired using a DigiGel-Digital Gel Image system (TOPBIO Ltd.), then analyzed and adjusted by ImageJ software (v1.53, NIH, Bethesda, MD, USA).

### 4.10. Aggregation Analysis

Overnight cultures of piliated GC were swabbed from a GCK plate and suspended in basal medium (DMEM: Ham F12 (1:1), 7% FBS); 2 × 10^6^ GC were incubated in each well of 8-well coverslip-bottom chambers (Sigma-Aldrich, St. Louis, MO, USA) for 6 h at 37 °C with 5% CO_2_ statically to allow the GC to form aggregates. Images were taken by a Zeiss Axio observer microscope, and the aggregation size was measured using ImageJ by circling and measuring the area of each aggregate.

### 4.11. Immunofluorescence Analysis of Polarized Epithelial Cells

Immunofluorescence analysis was performed using a previously published protocol [26]. After GC inoculation and incubation apically with polarized T84 cells, transwells were washed with PBS and fixed with paraformaldehyde, followed by permeabilization and staining with anti-E-cadherin (BD), anti-CEACAM1/3/6 (YTH71.3, Santa Cruz Biotechnology, Inc., Dallas, TX, USA), anti-GC antibodies [15], and Hoechst 33342 (Sigma-Aldrich, St. Louis, MO, USA). Images were acquired as z-series of 0.37 μm slices by ZEISS LSM 680 confocal microscopy. E-cadherin distribution was analyzed by measuring the fluorescent intensity ratio (FIR) at the apical junctional to the cytoplasmic area of over 30 individual cells using XY images as previously described [36]. The CEACAM-GC recruitment was analyzed using NIH ImageJ software, and the percentage of GC clusters with CEACAM patches in the vicinity was determined using 3-serial captured images.

### 4.12. Statistical Analysis

Statistical significance was assessed using the Student’s two-tail *t*-test or one-way analysis of variance (ANOVA) followed by Tukey’s multiple comparisons test depending on comparable properties. All the data were confirmed to fit into Gaussian distribution by the Shapiro–Wilk test for normality or lognormality. Homogeneity of variance was confirmed using Bartlett’s test and F-test. A value of *p* < 0.05 was considered statistically significant. All the analyses were performed using Prism9 software (GraphPad, San Diego, CA, USA).

## Figures and Tables

**Figure 1 pathogens-11-00217-f001:**
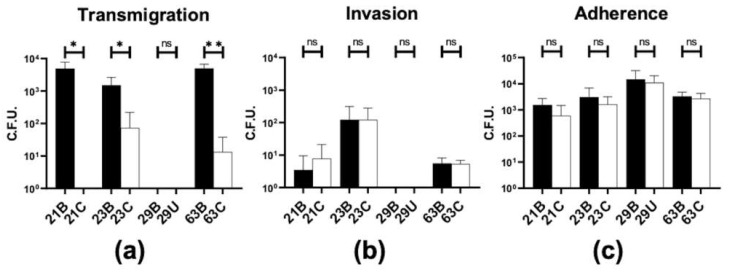
Transmigration, adherence, and invasion of isolates in polarized epithelial cells. Polarized T84 cells were apically inoculated with different GC isolates. The basal medium was collected to determine transmigrated GC (**a**). Invaded (**b**) and adhered GC (**c**) were quantified by gentamicin-resistant and adherence assays. Data were generated using three biological replicates with triplicate samples per replicate. Significance was determined using a Student’s *t*-test (mean ± SD, ns = *p* > 0.05, * *p* < 0.05, ** *p* < 0.01).

**Figure 2 pathogens-11-00217-f002:**
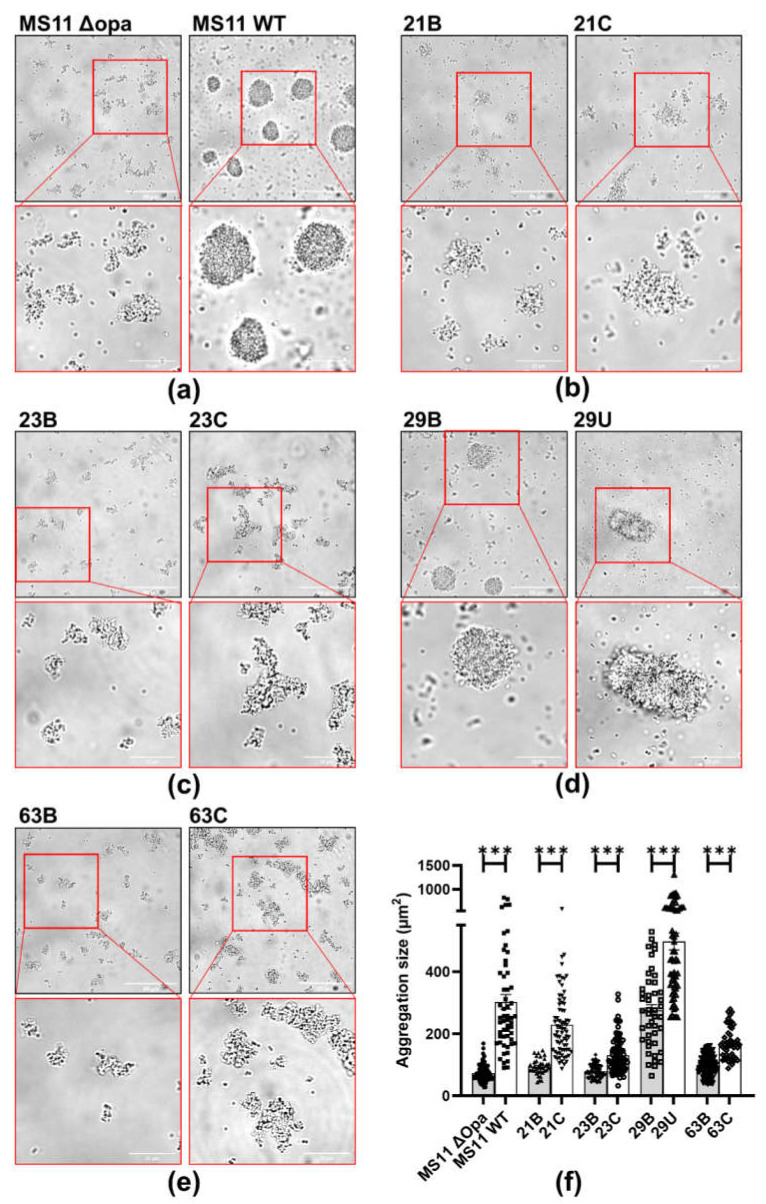
Aggregation of clinical isolates. Blood and cervical or ureteral isolates were seeded in coverslip-bottomed chamber, incubated for 6 h, and imaged by Zeiss Axio Observer microscope. (**a**–**e**) Representative images. (**f**) Sizes of aggregations were evaluated by measuring the average occupying area of individual aggregates in each image using NIH ImageJ. Scale bar: 50 μm (upper) and 20 μm (lower). The data were generated from eight randomly acquired fields from at least three independent experiments. Each data point indicates an individual aggregation. Statistical significance was determined using Student’s *t*-test (mean ± SEM, *** *p* < 0.001).

**Figure 3 pathogens-11-00217-f003:**
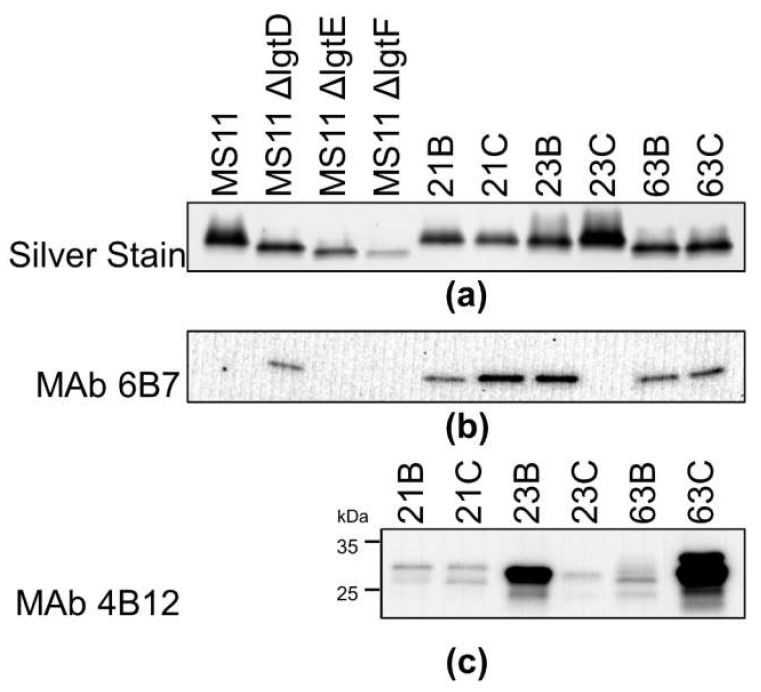
Characterization of LOS and Opa expression in localized and disseminated isolates. LOS extracted or whole-cell lysates from isolates were analyzed by SDS-PAGE and/or Western blotting. (**a**) LOS molecules were visualized by silver stain. (**b**) Immunoblot with MAb 6B7 was performed to detect the lacto-*N*-neotetraose structure of LOS. (**c**) Immunoblot with MAb 4B12 was performed to detect Opa proteins (i.e., loaded with 20 μg proteins in each lane and 60 μg in lane 63B). Shown are representative blots from three independent experiments.

**Figure 4 pathogens-11-00217-f004:**
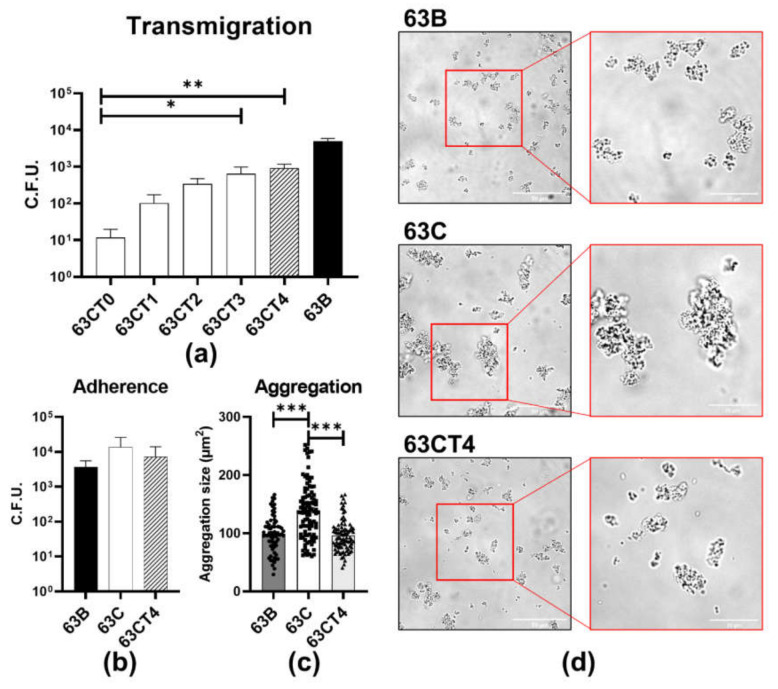
Transmigration, adherence, and aggregation of blood, cervical, and transmigration-enhanced cervical isolate. (**a**) Polarized T84 cells were apically inoculated with different generations of transmigration-enhanced cervical isolates. The basal medium was collected to determine transmigrated GC. (**b**) Blood, cervical, and transmigration-enhanced cervical isolate (the cervical isolates after 4 times transmigration, 63CT4) were incubated on the apical side of polarized T84 monolayers for 3 h and washed to determine adhered GC. (**c**,**d**) Blood, cervical, and transmigration-enhanced cervical isolate were cultured in coverslip-bottomed chambers at a density of 1 × 10^7^ cfu/mL (200 µL volume) and incubated for 6 h and imaged. The occupied areas of individual aggregates were measured using NIH ImageJ. Each data point indicates an individual aggregate. Scale bar: 50 μm (left) and 20 μm (right). The data represent the analysis of at least three randomly acquired images per condition from each of three independent experiments. Statistical significance was determined using a Student’s *t*-test (mean ± SEM, * *p* < 0.05, ** *p* < 0.01, *** *p* < 0.001).

**Figure 5 pathogens-11-00217-f005:**
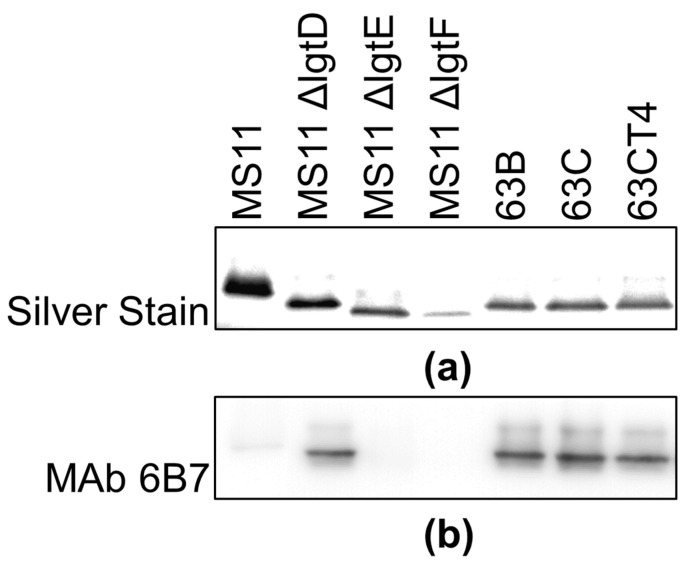
Characterization of LOS in transmigration-enhanced isolates. Crude LOS extracted from isolates was analyzed on an SDS-PAGE gel. (**a**) Silver stain of the gel was performed to visualize LOS. (**b**) Immunoblots of transferred LOS from gels were probed with MAb 6B7 for distinguishing different LOS structures.

**Figure 6 pathogens-11-00217-f006:**
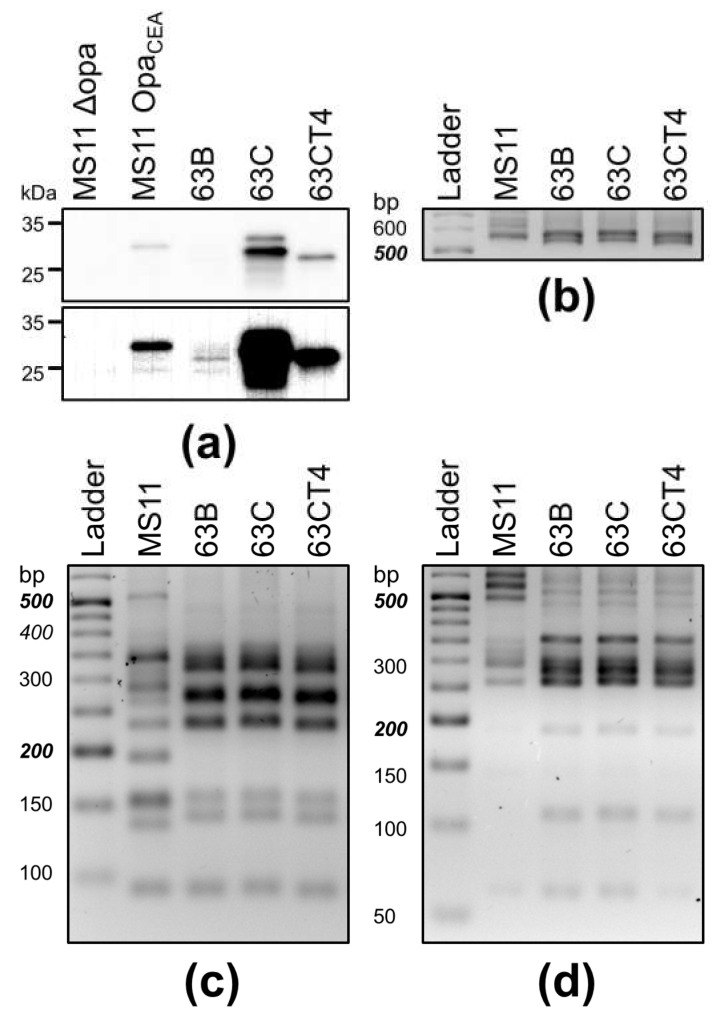
Variants and genomic organization of Opas expressed by isolates from patient 63 and the transmigration-enhanced cervical isolate 63CT4. (**a**) Isolate 63B, 63C, 63CT4, MS11ΔOpa, and Opa_CEA_ were lysed analyzed by Western blot probing with MAb 4B12. Equal amounts of whole-cell lysates individual isolates were analyzed. Shown are images with low contrast (upper panel) and high contrast (lower panel) of a representative blot. (**b**–**d**) Genomic DNA was isolated from each isolate. Amplification of Opa genes (**b**) was digested by TaqI (**c**) and Hpall (**d**).

**Figure 7 pathogens-11-00217-f007:**
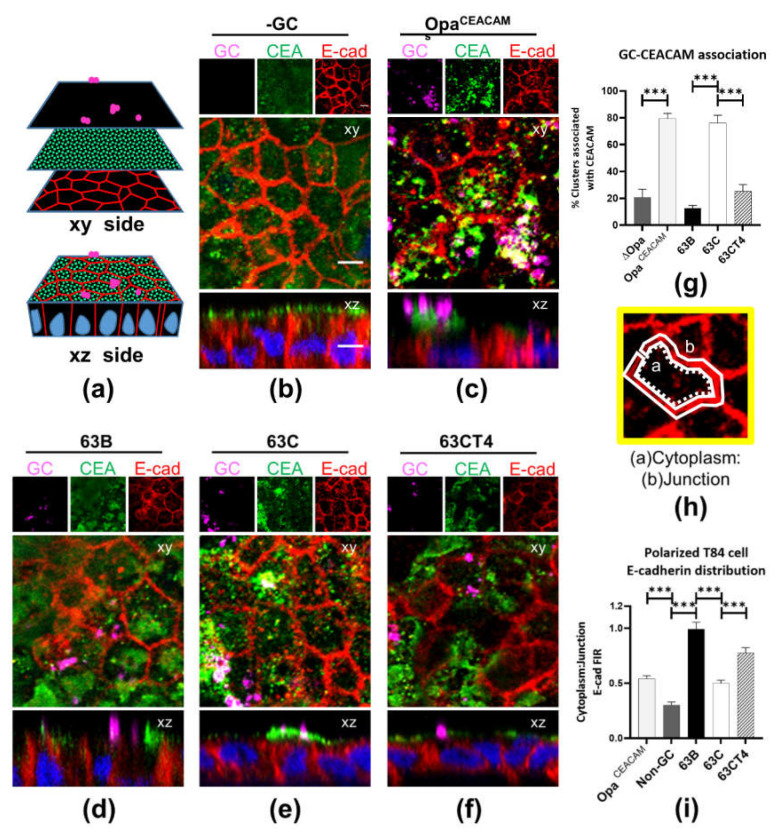
Effects of different GC isolates on the distribution of CEACAMs and the junctional protein E-cadherin in epithelial cells. Polarized T84 cells on transwells were inoculated with GC for 6 h. Cells were fixed, permeabilized, and stained for CEACAMs, the junctional protein E-cadherin, GC, and nucleus. Z-series images were acquired using a confocal microscope; 1 μm z-series layer at the apical surface were merged and quantified. (**a**) The illustration demonstrates a 3D analysis of GC, CEACAM, and E-cadherin distribution on the T84 cell surface. (**b**–**f**) Shown are representative xy (top and middle panels) and xz (bottom panels) images from no GC control (**b**), MS11 Opa_CEA_ (**c**), 63B (**d**), 63C (**e**), and 63CT4 (**f**) inoculated epithelial cells. Scale bar, 5 μm. (**g**) The percentage of GC aggregates with CEACAM patches in the vicinity was determined (MS11ΔOpa, which does not express Opa, was used as negative control). (**h**) Illustration of the fluorescence intensity ratio (FIR) of E-cad staining at the cytoplasmic to the cell-cell junctional region. (**i**) The average FIR was determined from > 30 cells of three individual experiments using NIH ImageJ. Statistical significance was determined using one-way ANOVA (Tukey’s test, mean ± SEM, *** *p* < 0.001).

**Table 1 pathogens-11-00217-t001:** MLST allele of housekeeping genes in clinically isolated *Neisseria gonorrhoeae* from different anatomical locations of the patients.

Patient	21		23		29		61		63	
Locus	Blood	Cervix	Blood	Cervix	Blood	Urethra	Blood	Cervix	Blood	Cervix
abcZ	129	129	200	200	126	126	126	126	128	128
adk	848	848	848	848	848	848	* 848	* 142	848	848
gdh	149	149	188	188	146	146	* 146	* 188	149	149
pdhC	901	901	940	940	1031	1031	901	901	153	153
pgm	839	839	839	839	839	839	* 839	* 981	65	65
aroE	810	810	810	810	810	810	810	810	810	810

* Difference of the housekeeping gene sequence between isolates from the same patient.

## Data Availability

Data of this study is available upon request to the authors.

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
