# Peer review of "In Vitro Analysis of Matched Isolates from Localized and Disseminated Gonococcal Infections Suggests That Opa Expression Impacts Clinical Outcome"

_pathogens, 2022, doi:10.3390/pathogens11020217_

Round 1
Reviewer 1 Report
A brief summary The authors demonstrated a critical role of Opa phase variation in determining the clinical outcomes of Neisseria gonorrhoeae infection. Using patient isolates, this study provides evidence that N. gonorrhoeae can switch infection modes from local into dissemination by changing Opa expression.
General concept comments The manuscript is clear, scientifically well concepted and experimental design and methods are well described. Figures and tables are appropriate and are showing the data properly. The conclusion is coherent and drawn from the data. References are adequately cited.
Specific comments Please, if the authors can refer in the Discussion section to possibility of use the results of this study in the future treatment or prevention of gonococcal disseminated infections.
Author Response
Specific comments Please, if the authors can refer in the Discussion section to possibility of use the results of this study in the future treatment or prevention of gonococcal disseminated infections.
We thank reviewer’s suggestion and have added a sentence to state this work’s potential clinical application (Line 379-381).
Reviewer 2 Report
The paper from Wu et al., is a very nicely written study of Opa expression in gonococcal isolates from the cervix and blood of individual patients and how expression of Opa and LOS impact on transmigration of epithelial cells in vitro. The title of the paper might seem a bit brief and perhaps needs amending to reflect the limited number of patients studied, and the fact that a lot of the data are from in vitro studies.
There is a groaning literature on Opa expression and clinical outcome, for example from many a human challenge study. The paper adds novelty to the role of Opa in directing invasion or transmigration. The authors should make clear that transmigration is an intercellular event, which to the un-initiated reader is difficult to comprehend.
Minor Comments:
- I think a Table that describes the phenotypes of the isolates and other strains used in the paper would be useful at the beginning of the results, i.e. to clarify Opa, LOS and pilus and PorB expression.
- In Figure 1C, between 29B and 29U, is there supposed to be a column or is it empty? There is a vertical line in my downloaded proof.
- Figure 2. Aggregation - are the differences in aggregation influenced at all by the piliation status of these isolates? Pilus expression is known to cause bundling and aggregation, so I wonder what happens if you add non-piliated variants of your isolates into your system (if indeed the parents are piliated? See comment 1).
- Fig 7g. Is the deltaOpa variant in the figure the MS11 Opa-non-expressor? It is not clear in legend.
- Limitation of the study - could be the use of the carcinoma cell line T84. I know it has been used in other studies and by other groups looking at tight junctional interactions, and it has the advantage of polarity, but perhaps the authors ought to accept that using primary cells of the urogenital tract might be more appropriate. A comment in the discussion would be appropriate.
Author Response
We greatly appreciate the reviewer for recognizing the merit of our work. We understand that this work is just the beginning of finding the mechanistic explanation of how GC switch between localized/disseminated infections from the clinical perspective, and some limitations need to be stated in the context as reviewers commented. We believe that we can address all the comments within this manuscript and will be ready for publication.
The paper from Wu et al., is a very nicely written study of Opa expression in gonococcal isolates from the cervix and blood of individual patients and how expression of Opa and LOS impact on transmigration of epithelial cells in vitro. The title of the paper might seem a bit brief and perhaps needs amending to reflect the limited number of patients studied, and the fact that a lot of the data are from in vitro studies.
We thank reviewer’s suggestion and have modified the title (Line 2-4).
There is a groaning literature on Opa expression and clinical outcome, for example from many a human challenge study. The paper adds novelty to the role of Opa in directing invasion or transmigration. The authors should make clear that transmigration is an intercellular event, which to the un-initiated reader is difficult to comprehend.
We thank reviewer’s suggestion. We have added a sentence to clarify the increase of intercellular transmigration in disseminated isolates (Line 122).
Minor Comments:
- I think a Table that describes the phenotypes of the isolates and other strains used in the paper would be useful at the beginning of the results, i.e. to clarify Opa, LOS and pilus and PorB expression.
We thank reviewer’s suggestion but unfortunately, the most promising phenotypic characterization of a GC colony under dissecting microscope is the presence/absence of pilus based on colony color and size (Swanson, J., Kraus, S. J., & Gotschlich, E. C. (1971). Studies on gonococcus infection: I. Pili and zones of adhesion: Their relation to gonococcal growth patterns. The Journal of experimental medicine, 134(4), 886-906.). We have used this characterization to streak and use pilus-positive colonies in our study. The phenotypical characterization of Opa and LOS expression were done by western blot in Figure 3 & 7 and constitutively-expressed PorB was genotypically characterized in Table S1.
- In Figure 1C, between 29B and 29U, is there supposed to be a column or is it empty? There is a vertical line in my downloaded proof.
We have corrected the figure 1C accordingly.
- Figure 2. Aggregation - are the differences in aggregation influenced at all by the piliation status of these isolates? Pilus expression is known to cause bundling and aggregation, so I wonder what happens if you add non-piliated variants of your isolates into your system (if indeed the parents are piliated? See comment 1).
We thank reviewer for asking this question. The pilus expression can indeed contribute to the GC-GC interaction and aggregation (Wang, L. C., Litwin, M., Sahiholnasab, Z., Song, W., & Stein, D. C. (2018). Neisseria gonorrhoeae aggregation reduces its ceftriaxone susceptibility. Antibiotics, 7(2), 48.). The lack of pilus was shown significantly decreased aggregation size. However, clinically, the pilus is needed for adherence to the epithelial cells and initiation of infection. Thus, we have used pilus-positive colonies in this work to best represent the infection.
- Fig 7g. Is the deltaOpa variant in the figure the MS11 Opa-non-expressor? It is not clear in legend.
Yes, the ΔOpa in the figure 7g is the MS11 Opa-non-expressor. We have added a sentence in the figure 7g accordingly (Line 295-296).
- Limitation of the study - could be the use of the carcinoma cell line T84. I know it has been used in other studies and by other groups looking at tight junctional interactions, and it has the advantage of polarity, but perhaps the authors ought to accept that using primary cells of the urogenital tract might be more appropriate. A comment in the discussion would be appropriate.
We agree with the reviewer’s concern and have added a sentence in the discussion to state the future work can be done with primary cell line and tissue explant as well (Line 376-377).